# Strength Through Diversity:
# Robust Behavior Learning via Mixture Policies

**Tim Seyde**[1]
MIT CSAIL

**Wilko Schwarting**
MIT CSAIL

**Igor Gilitschenski**
University of Toronto

**Markus Wulfmeier**[*]
DeepMind

**Daniela Rus**[*]
MIT CSAIL

**Abstract:** Efficiency in robot learning is highly dependent on hyperparameters. Robot morphology and task structure differ widely and finding the optimal setting typically requires sequential or parallel repetition of experiments, strongly increasing the interaction count. We propose a training method that only relies on a single trial by enabling agents to select and combine controller designs conditioned on the task. Our Hyperparameter Mixture Policies (HMPs) feature diverse sub-policies that vary in distribution types and parameterization, reducing the impact of design choices and unlocking synergies between low-level components. We demonstrate strong performance on continuous control tasks, including a simulated ANYmal robot, showing that HMPs yield robust, data-efficient learning. [2]

**Keywords:** Learning Control, Hierarchical Optimization, Sample Efficiency

## 1 Introduction

Real-world autonomous robots require versatile controllers that continuously adapt behavior to changing environmental conditions and task specifications. Reinforcement learning (RL) has driven success in modeling complex control strategies in games [1, 2], simulated robotics [3] and real-world systems [4, 5]. However, efficient learning is often conditioned on good parameter selection and may require tuning for each task or domain [6]. Common approaches to hyperparameter optimization leverage parallel or sequential evaluation strategies. While parallel strategies are computationally costly and need not improve on random search [7], sequential strategies [8, 9] are time-intensive with hybrid approaches trading-off one for the other [10]. Gradient-based methods enable online adaptation at the cost of requiring differentiable objectives. Selecting parameters efficiently is essential to learning capable robot controllers.

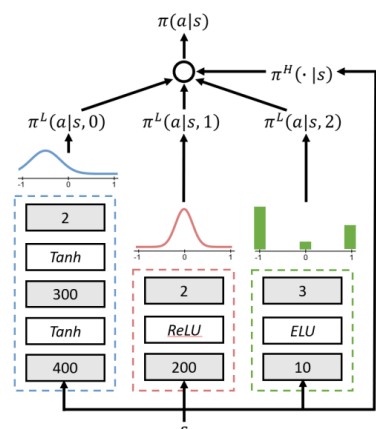

Figure 1: Hyperparameter Mixture Policy (HMP) with diverse low-level distributions. The agent can seamlessly adapt its policy structure to the presented tasks by modulating component activations in a single trial.

The nature of motion planning problems further dictates suitable controller designs for learning. Continuous control can represent intricate transitions in state-action space to yield highly-optimized behaviors through local exploration or generate smooth references for a low-level tracking controller. Discrete control can leverage reduced resolution for coarse exploration and readily encodes

---

[1]Correspondence to `tseyde@mit.edu`. [*]Equal advising.

[2]Please find additional details at `https://sites.google.com/view/diversity2021`

bang-bang responses to switching dynamics. Optimal controller selection does not have to be uni-modal and can vary with different phases of a task and stages of the learning process. It may then be advantageous to provide agents with a diverse set of controllers that differ in their parameterization. This enables agents to select designs suitable for the presented task and unlock compositional synergies. In order to support this type of compositionality, we extend the perspective of previous work on hierarchical reinforcement learning [11].

In this paper, we propose a hierarchical policy over a diverse mixture of low-level controllers to improve robustness and reduce the necessity for parameter tuning and related data requirements. The low-level controllers are diverse in architecture, hyperparameters, and distribution characteristics to provide the robot with a rich set of controller designs. Our approach then enables learning robots:

- to optimize a set of diverse controller designs concurrently for increased data-efficiency,
- to self-select suitable controllers conditioned on the task for reduced human parameter tuning,
- to compose behaviors from multiple controller designs for exploitation of emergent synergies.

We evaluate performance on a variety of torque-control tasks from the DeepMind Control Suite [12]. Additionally, we investigate learning of PD-control targets for the ANYmal robot in RaiSim [13], which was the foundation for Sim2Real transfer in Lee et al. [4]. Throughout, we demonstrate that enabling agents to operate over a diverse set of controllers guards against individual failure modes and unlocks synergies between different controller designs. While the high-level selector introduces its own hyperparameters, we demonstrate its robustness to loss of state information by modelling unconditional component selection and subjecting the selector to adversarial distractor components.

## 2 Related Work

The performance of deep RL algorithms is strongly tied to hyperparameter choices [6, 14, 15]. Commonly, hyperparameter optimization is performed in multiple experiments via sets of agents or tasks [16]. Simple parallel strategies include expert selection or grid-search [17, 18], and can be less efficient than random search [7]. Bayesian Optimization provides more structure at the cost of sequential evaluations [9, 19, 20]. Evolutionary strategies [8, 21, 22] enable discontinuous optimization and can evolve parameters at different rates, but typically use sequential evaluation. Population-Based Training (PBT) [10] alternatively evolves parameter variations online in parallel. This requires populations of agents each with their own environment, leading to significant data and computational requirements. Our work is also related to neural architecture search [23]. In particular, similarities can be found to methods that render architecture search differential [24, 25], enabling direct optimization of the effectively-used architecture during a single experiment.

Optimizing hyperparameters during the lifetime of a single agent reduces these requirements. Gradient-based optimization ([26, 27]) yields online adaptation when the objective is differentiable in the parameters. HOOF [28] extends towards non-differential aspects and enables gradient-free off-policy training with hyperparameter schedules. However, the optimized parameters need to directly affect the policy update, precluding e.g. application to architecture search. Related methods have been used to learn architecture schedules [29]. With HMPs, we consider a single agent lifetime to reduce data and computational requirements. We consider policies that vary in their parameters, architecture, and distributional characteristic. Formulating a mixture over these diverse components allows us to evolve multiple controllers in parallel while the agent modulates activation based on expected performance. The approach further enables combining controllers with different parameters.

The problem of hyperparameter choice can further be framed as a contextual bandit problem [30]. In comparison to mixture agents, this formulation does not natively share data across policies with different parameters. Mixtures of specialized motion controllers naturally arise in form of motor primitives [31, 32]. The application of dynamical movement primitives [33] to robot control via a mixture library has been shown in [34]. Mixture distributions have a long-standing history to model diverse and multi-modal data [35]. In RL, they have been applied as a mixture of linear Gaussians in which component specialization is achieved by introducing entropy bounds [36] and provide

an inference perspective to the options framework which models an agent via the separation into high-level controller and low-level behaviours [11, 37, 38]. Similarly, quality diversity algorithms evolve repertoires of diverse low-level skills which a high-level selector may act on [39, 40, 41]. We build on the training of mixture policies via weighted maximum likelihood optimization, which has previously been used in connection with information asymmetry to generate diverse, compositional skills [42]. Diversity in mixture policies has been explored to strengthen skill discovery by explicitly optimizing for diversity-related objectives [43, 44, 45, 46] with fixed architectures and a single set of hyperparameters.

## 3 Preliminaries

We formulate controller optimization as a Markov Decision Process (MDP) defined by the tuple $\{\mathcal{S}, \mathcal{A}, \mathcal{T}, \mathcal{R}, \gamma\}$, where $\mathcal{S}$ and $\mathcal{A}$ denote the state and action space, respectively, $\mathcal{T} : \mathcal{S} \times \mathcal{A} \to \mathcal{S}$ represents the transition distribution, $\mathcal{R} : \mathcal{S} \times \mathcal{A} \to \mathbb{R}$ the reward mapping, and $\gamma \in [0, 1)$ the discount factor. We define $s_t$ and $a_t$ to be the state and action at time $t$, respectively. Let $\pi_\theta(a|s)$ denote a policy distribution with parameters $\theta$ and define the discounted infinite horizon return $G_t = \sum_{t'=t}^{\infty} \gamma^{t'-t} R(s_{t'}, a_{t'})$, where $s_{t+1} \sim \mathcal{T}(s_{t+1}|s_t, a_t)$ and $a_t \sim \pi_\theta(a_t|s_t)$. Our goal is to learn the optimal policy maximizing $G_t$ under unknown dynamics and reward mappings. This is typically done by modeling $\pi_\theta(a_t|s_t)$ as a Gaussian distribution with a neural network predicting the mean and variance from $s_t$. In this work, we consider a more diverse class of policy distributions.

## 4 Hyperparameter Mixture Policies

We propose Hyperparameter Mixture Policies (HMP) to train a hierarchical policy over diverse low-level controllers with distinct hyperparameters and distribution characteristics. The resulting mixture is given by

$$\pi_\theta(a|s) = \sum_{i=1}^{M} \pi_\theta^H(i|s) \pi_\theta^L(a|s, i), \quad (1)$$

with $\pi^H(i|s)$ and $\pi^L(a|s, i)$ as the weight and probability density of component $i$. Thus, $\pi^H$ is a high-level selector and $\pi^L$ a sub-policy from a diverse set of controller designs. The agent then self-selects the most suitable controller for individual phases of a task or stages of the learning process. This enables robust adaptation to a broad range of motion planning problems while reducing the necessity of manual parameter tuning and sequential experiment design.

---

**Algorithm 1:** Hyperparameter Mixture Policies

**Initialize:** $N_{\text{step/target}}$: (target) update steps, $N_s$: action samples per state, $\epsilon$: KL bounds

**while** $k \leq N_{step}$ **do**

  **for** $k \leftarrow 1$ **to** $N_{target}$ **do**

    Sample batch of trajectories $\tau$ from memory $B$, $N_s$ actions from $\pi_{\theta_k}$ to estimate expectations

    Compute gradients over batch for $\pi_\theta, \eta, \lambda_p, Q_\phi$

$$\delta_\pi \leftarrow -\nabla_\theta \sum_{s \sim \mathcal{D}} \sum_{j=1}^{N_s} \left[ \exp\left(\frac{Q(s, a_j)}{\eta}\right) \right.$$
$$\left. \cdot \log \pi_\theta(a_j \mid s) \right]$$
$$+ \sum_p \lambda_p \left( \epsilon_p - \mathbb{E}_{s \sim \mathcal{D}} \left[ \text{KL}(\pi_\theta || \pi_{\theta_k}) \right] \right) \quad (8)$$

$$\delta_\eta \leftarrow \nabla_\eta g(\eta) = \nabla_\eta \eta \epsilon + \eta \sum_{s \sim \mathcal{D}} \log \frac{1}{N_s}$$
$$\sum_{j=1}^{N_s} \left[ \exp\left(\frac{Q(s, a_j)}{\eta}\right) \right] \quad (5)$$

$$\delta_{\lambda_p} \leftarrow \nabla_{\lambda_p} \lambda_p \left( \epsilon_p - \mathbb{E}_{s \sim \mathcal{D}} \left[ \text{KL}(\pi_\theta || \pi_{\theta_k}) \right] \right) \quad (8)$$

$$\delta_Q \leftarrow \nabla_\phi \sum_{(s,a) \sim \mathcal{D}} (Q_\phi(s, a) - Q^{\text{ret}})^2 \quad (10)$$

    Apply gradients to update $\pi_{\theta_{k+1}}, \eta, \lambda_p, Q_\phi$

  Update target networks $\pi'_\theta = \pi_\theta, Q'_\phi = Q_\phi$

---

### 4.1 Policy Improvement

We use an actor-critic algorithm where policy improvement relies on two stages as in [47]. First, a non-parametric policy $q(a|s)$ is optimized on samples from the state-action value function $Q^\pi$ under the constraint of remaining close in expectation to the current parametric policy $\pi_\theta$. The parametric policy is then updated to better approximate the non-parametric target. By performing the actual policy improvement with a non-parametric policy, we bypass the need for gradient estimation via likelihood ratio [48] or reparametrization trick [49]. In addition, this perspective enables the optimisation of mixture distributions in reinforcement learning without continuous relaxation [50].

**Step 1 - Fitting the Non-parametric Policy**

As we do not have access to the ground-truth state-action value function $Q$ we employ a learned approximation $Q_\phi$, parameterized by $\phi$, and formulate the objective at iteration $k$ as

$$\max_q J(q) = \mathbb{E}_{q,s\sim D}\left[Q_\phi(s,a)\right], \tag{2}$$

$$\text{s.t. } \mathbb{E}_{s\sim D}\left[\text{KL}(q(a|s)||\pi_{\theta_k}(a|s))\right] \leq \epsilon, \tag{3}$$

where $\epsilon$ defines a trust-region around the current parametric policy, $\pi_{\theta_k}$. This can be solved to provide a closed-form relation in terms of the current parametric policy

$$q_k(a|s) \propto \pi_{\theta_k}(a|s)\exp\left(\frac{Q_\phi(s,a)}{\eta}\right), \tag{4}$$

where $\eta$ is computed by minimizing the dual function

$$g(\eta) = \eta\epsilon + \eta\mathbb{E}_{s\sim\mathcal{D}}\left[\log\int\pi_{\theta_k}(a|s)\exp\left(\frac{Q_\phi(s,a)}{\eta}\right)\mathrm{d}a\right]. \tag{5}$$

**Step 2 - Updating the Parametric Policy**

The parametric policy $\pi_\theta$ is optimized to approximate the non-parametric policy $q$ by minimizing their KL divergence as

$$\min_\theta L(\theta) = \mathbb{E}_{s\sim\mathcal{D}}\left[\text{KL}(q_k(a|s)||\pi_\theta(a|s))\right]. \tag{6}$$

Plugging in (4) and introducing an additional KL constraint to enable generalization beyond the sample distribution yields

$$\max_\theta J(\theta) = \mathbb{E}_{s\sim\mathcal{D}}\left[\mathbb{E}_{\pi_{\theta_k}}\left[\exp\left(\frac{Q_\phi(s,a)}{\eta}\right)\log\pi_\theta(a|s)\right]\right],$$
$$\text{s.t. } \mathbb{E}_{s\sim\mathcal{D}}\left[\text{KL}(\pi_{\theta_k}(a|s)||\pi_\theta(a|s))\right] \tag{7}$$

The update proceeds via a Lagrangian relaxation of the KL constraint enabling the application of gradient-based optimization. We further decouple the KL constraint and define independent constraints for each distributional parameter $p$ in both the high-level selector and low-level control policies [47, 42]. We define these separately per diverse component to accommodate differences in the distributional parameter dynamics between low-level controllers (in comparison to e.g. Wulfmeier et al. [42]). To enable training of diverse distributions with different constraints, we enforce the KL constraints per component. Additional changes to enable training of mixed continuous discrete policies are described in Section 4.3. We obtain updated parameters $\theta_{k+1}$ as a solution of

$$\max_\theta\min_{\lambda_p>0} L(\theta,\lambda_p) = \mathbb{E}_{s\sim\mathcal{D},\pi_{\theta_k}}\left[\exp\left(\frac{Q_\phi(s,a)}{\eta}\right)\cdot\log\pi_\theta(a|s)\right]$$
$$+ \sum_p\lambda_p\left(\epsilon_p - \mathbb{E}_{s\sim\mathcal{D}}\left[\text{KL}(\pi_\theta(a|s)||\pi_{\theta_k}(a|s))\right]\right) \tag{8}$$

where we sum over decoupled components, each only varying along its respective parameter $p$. We also introduce component specific Lagrangian multipliers $\lambda_p$ and KL bounds $\epsilon_p$. A two component mixture of e.g. a Categorical ($\alpha_1$) and a Gaussian ($\mu_2, \Sigma_2$) would then yield $p = \{\alpha_{\text{HL}}, \alpha_1, \mu_2, \Sigma_2\}$.

### 4.2 Policy Evaluation

In order to stabilize off-policy learning of the state-action value function $Q_\phi$ we leverage the Retrace algorithm [51]. Here, we truncate the infinite series after 10 steps and bootstrap from the target state-action value network, with details provided in Appendix D. To increase efficiency, we further consider two-step transitions by squashing consecutive timesteps before adding them to memory.

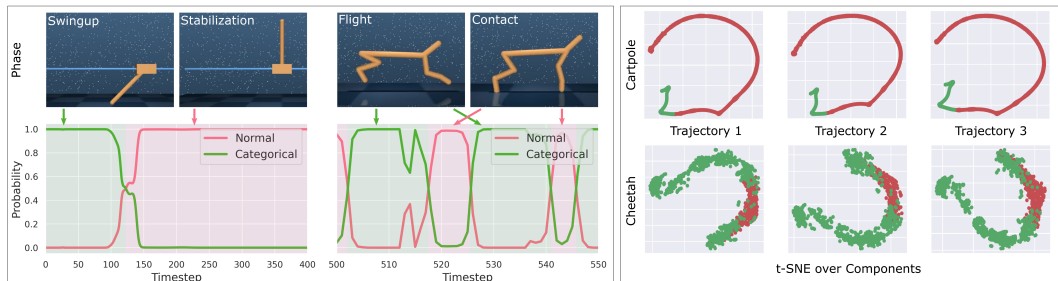

Figure 2: Component specialization within an HMP consisting of a narrow Gaussian ($\sigma_{\text{ini}} = 0.3$) and a coarse Categorical ($n_{\text{bin}} = 2$). On Cartpole, bang-bang control enables fast swing-up and fine-grained control stabilizes the upright. On Cheetah, fine-grained control coordinates the contact phase and bang-bang control retracts the limbs during flight phase. Providing an agent with diverse low-level controllers can unlock synergistic specialization that is consistent across states (t-SNE).

### 4.3 Combining Continuous and Discrete Distributions

Continuous and discrete policies do not share the same support. Furthermore, actions are subject to numerical cut-off errors. In practice this can result in the action-likelihood of discrete policies being 0 most of the time. To facilitate training with diverse mixtures, we approximate discrete components by piece-wise constant pseudo-densities for backpropagation. Thus, out-of-distribution samples are mapped into the support for probability computation. For query action $a$ and a discrete mixture component $i$ with finite support $\mathcal{C}_i$ we obtain the corresponding piece-wise constant pseudo-density

$$\pi_\theta^L(a|s,i) = \sum_{\tilde{a} \in \mathcal{C}_i} p_i(\tilde{a}|s) \cdot \mathbb{1}_{B_\delta(\tilde{a})}(a) \, , \tag{9}$$

where $p_i(\tilde{a}|s)$ is the probability of $\tilde{a}$ in the original discrete distribution, $\mathbb{1}$ is the indicator function, and $B_\delta(\tilde{a})$ is a ball of radius $\delta$ around $\tilde{a}$ (we use $\delta = 0.1$ here). This improves sharing of gradient information between continuous and discrete policies and enables discrete components to train on samples generated by continuous components to accelerate learning.

## 5 Experiments

We benchmark the performance of HMPs on continuous control in the DeepMind Control Suite [12], learning PD-control for ANYmal in RaiSim [52, 13], and manipulation tasks in Metaworld [53]. We further compare to the gradient-free hyperparameter optimizer HOOF [28] in OpenAI Gym [54]. To better isolate the effects of diversity, we consider a standard Gaussian policy and evaluate single parameter variations together with their composition into diverse HMPs. We furthermore investigate HMPs consisting of randomly sampled components and show that strong performance can be recovered. Overall, we find diversity to enable robust learning across tasks and to guard against failure modes of individual components. Our figures visualize performance mean and standard deviation.

### 5.1 Qualitative Example

We provide an illustrative example of component specialization within a diverse policy in Figure 2. The agent combines a localized Gaussian ($\sigma_{\text{ini}} = 0.3$) with a coarse Categorical ($n_{\text{bin}} = 2$) policy. On a Cartpole swing-up task, the agent leverages bang-bang control for swing-up and continuous control for stabilization. On a Cheetah locomotion task, continuous control coordinates the intricate contact phase and bang-bang control quickly retracts the limbs during the flight phase. Applying t-SNE dimensionality reduction yields consistent clustering across trajectories, indicating consistent component specialization. This aligns with human intuition and highlights the promise of composition, further motivating HMPs and analysis of synergies in heterogeneous mixture policies.

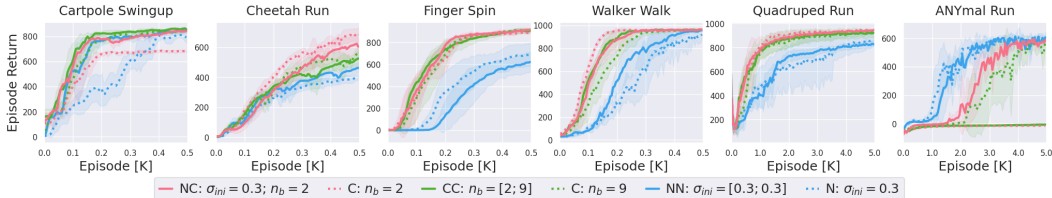

Figure 3: Combining continuous and discrete distributions to unlock synergies. Pairing a narrow Gaussian with a bang-bang controller yields strong performance, guarding against component failure (bang-bang on ANYmal, Gaussian on Finger) and improving on individual performance (HMP on Cartpole). Coarse control can drive exploration, fine-grained control can enable accurate tracking.

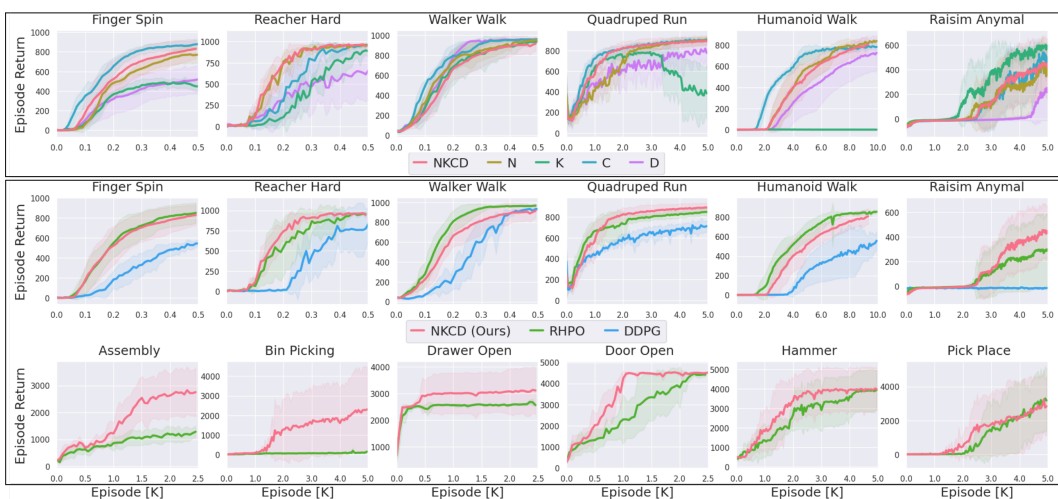

Figure 4: Top - HMP consisting of a Gaussian (N), Kumaraswamy (K), Categorical (C) and Discrete Gaussian (D) heads with individual components for reference. Bottom - HMP instance compared to baselines RHPO and DDPG. Our HMP is robust to sub-policy failure (e.g. K on Humanoid) and yields strong performance especially on the real-world inspired ANYmal and manipulation domains.

## 5.2 Heterogeneous Distributions

**Compositional Solutions**  We further evaluate synergies arising from heterogeneous mixtures by combining a narrow Gaussian policy ($\sigma_{\text{ini}} = 0.3$) with a Categorical policy (with $n_{\text{bin}} \in \{2, 9\}$). Figure 3 indicates that performance of the Gaussian significantly improves in combination with a bang-bang policy ($n_{\text{bin}} = 2$) for torque-control (panels 1-5) (see also [55]). Conversely, the Gaussian guards against the failure mode induced by bang-bang control on ANYmal (panel 6). We can further replace the Gaussian with a more fine-grained Categorical ($n_{\text{bin}} = 9$) to reach comparable performance on the Control Suite tasks. However, the resulting mixture cannot compensate for bang-bang signals on ANYmal as discrete control is not well-suited for position reference generation.

**Diverse Distributions**  We broaden our analysis of diverse distributions and consider a Gaussian ($\sigma_{\text{ini}} = 1.0$), Kumaraswamy ($c_{\text{ini}} = 1.0$), Categorical ($n_{\text{bin}} = 5$) and Discrete Gaussian ($n_{\text{bin}} = 5$), as well as their combination into an HMP which we refer to as NKCD. Figure 4 highlights that the HMP is able to solve all tasks, guarding against individual component failure (e.g. K on Quadruped) or premature convergence (e.g. D on Reacher). On ANYmal, the HMP leverages the Kumaraswamy policy to outperform the Gaussian policy, while the Kumaraswamy is suppressed on the Humanoid to reach strong performance. This underlines the robust performance that diverse HMPs provide by evaluating multiple controller designs jointly, reducing environment-specific tuning and guarding from component failure. We compare the NKCD HMP to the RHPO [42] and DDPG [56] agents. RHPO leverages a homogeneous mixture policy consisting of 4 MPO-type Gaussians ($\sigma_{\text{ini}} = 1.0$).

Figure 4 shows that the HMP and RHPO outperform DDPG on all tasks. The HMP performs competitively with RHPO throughout and outperforms RHPO on the real-world inspired ANYmal and manipulation domains. This underlines HMP's ability to enable data-efficient learning by training multiple policy designs in parallel and transferring problem-specific controller selection to the agent.

**Random components**   We consider sampling sub-policies with random hyperparameters. This includes randomizing distribution type, initialization, architecture and activations of each component. Figure 5 provides performance statistics across 10 random instances of an HMP with 10 random components. We observe that random selection yields performance competitive with the optimized RHPO agent. Beyond random selection, the engineer may restrict the space of available sub-policies to inject structural priors into the learning process.

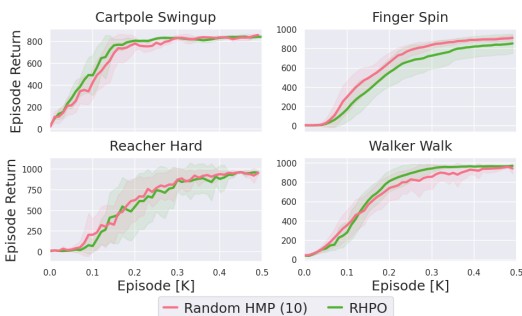

Figure 5: Random sub-policy parameterizations.

**Gradient-free optimization**   We compare the NKCD HMP to HOOF [28], which introduced a method for gradient-free hyperparameter optimization and evaluated on OpenAI Gym [54]. Their results are provided in Figure 6 for reference. We note that our diverse mixture displays competitive performance on these benchmarks without any fine-tuning for Gym domains.

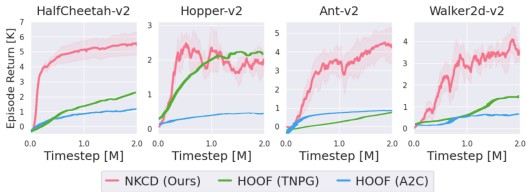

Figure 6: HMP vs. HOOF [28] on OpenAI Gym.

## 5.3   Hyperparameter Variations

In the following, we evaluate performance of different hyperparameter combinations on continuous control tasks from the DeepMind Control Suite and locomotion on the ANYmal robot in RaiSim. We consider a standard MPO parameterization as the baseline and analyze the impact of combining potentially sub-optimal parameter variations. To account for increased model capacity in mixture policies, we compare to a RHPO-type homogeneous mixture with standard MPO parameters.

**Initialization**   We vary the initial standard deviation of Gaussian policy heads as this can significantly impact the rate of convergence. The bounded action space of the agent is $a \in [-1.0, +1.0]^{|\mathcal{A}|}$. We consider the initial values $\sigma_{\text{ini}} = \{0.3, 1.0, 3.0\}$ with the standard literature value $\sigma_{\text{ini}}^{\text{s}} = 1.0$. Figure 7 (row 1) indicates that the Control Suite tasks favor large variance to drive exploration, while generating position targets on ANYmal requires low variance to avoid instability of the PD controller and subsequent falling. Generally, we find that a diverse policy improves performance over the weaker component, yielding a robust controller that can prevent individual failure modes. This is evident for ANYmal Run, where the high variance policy fails but the diverse mixture succeeds.

**Architecture**   We vary the layer structure with $\pi_l \in \{[20], [200], [200, 200]\}$ and standard value $\pi_l^{\text{s}} = [200]$. Figure 7 (row 2) indicates that performance is robust to architecture variations with increased capacity slightly improving performance. The heterogeneous mixture performs slightly better on the Control Suite tasks while the homogeneous mixture is slightly better on ANYmal Run.

**Activations**   We vary the policy activations with $\pi_a = \{\text{ELU, Leaky ReLU, Tanh}\}$ and $\pi_a^{\text{s}} = ELU$. Figure 7 (row 3) shows that the heterogeneous mixture improves performance over both the homogeneous mixture and the individual components on the Control Suite tasks. On ANYmal, combining Leaky ReLU and Tanh activations initially causes quick episode termination and delayed learning. Based on individual performance this is surprising and could indicate that ELU activations are better suited for composition on this task. This is the only instance where we observe reduced performance.

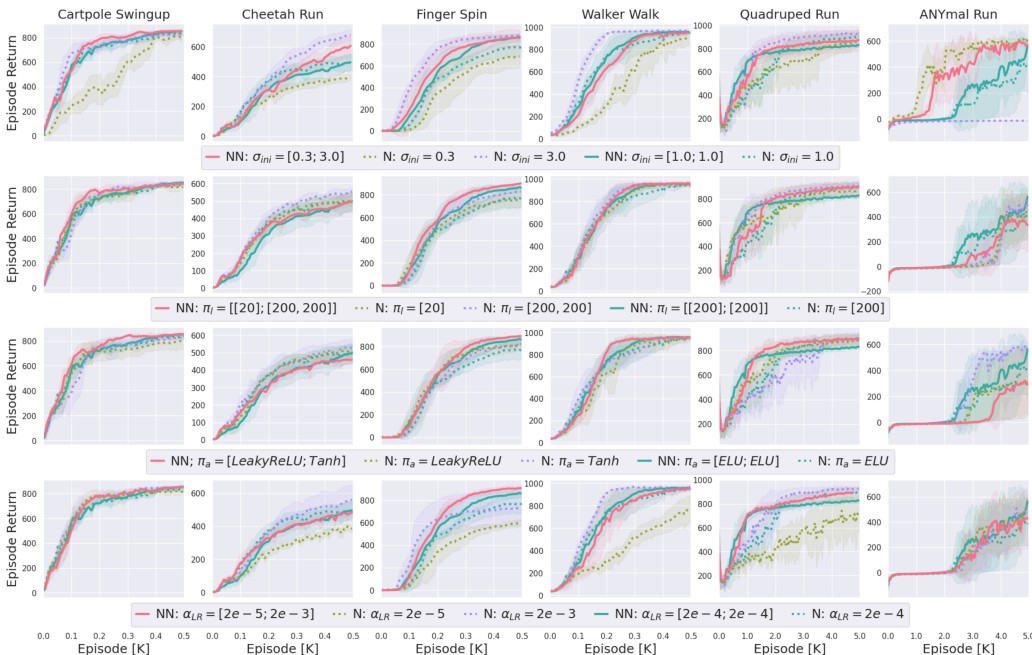

Figure 7: Performance of a diverse mixture (solid red), a homogeneous mixture (solid green), and individual components. The homogeneous mixture uses MPO parameters, while the diverse mixture combines potentially sub-optimal parameter variations. Generally, the diverse mixture performs competitively while guarding against sub-policy failure modes (e.g. $\sigma_{\text{ini}} = 3.0$ on ANYmal, top).

**Learning Rate**   We vary the policy learning rates such that $\alpha_{\text{LR}} \in 2 \times \{10^{-5}, 10^{-4}, 10^{-3}\}$ and $\alpha_{\text{LR}}^{\text{s}} = 2 \times 10^{-4}$. Figure 7 (row 4) indicates that smaller learning rates reduce efficiency. However, pairing a fast with a slow head yields competitive performance, significantly improving over the individual slow component, and outperforms a nominal mixture on the Finger and Quadruped tasks.

## 6   Conclusion

Finding the right hyperparameters has a considerable impact on performance when enabling robots to learn complex behaviors through interaction and recent progress in machine learning can often be traced back to better hyperparameter settings [57]. A sub-optimal algorithm with thoughtfully tuned hyperparameters easily outperforms a state-of-the-art approach that has not been tuned sufficiently. Tuning requires both domain knowledge and experience with the underlying algorithm. Even then, it still incurs a considerable computational cost that is particularly limiting when relying on real-world data. Our work proposes the use of diverse mixture policies to effectively mitigate this challenge. Moreover, we demonstrate the benefits of combining different distribution types and policy parameterizations from a perspective of compositionality in skill learning. Our Hyperparameter Mixture Policies (HMPs) induce diversity that can help in component specialization during different phases of a task, e.g. where certain movements require either coarse or fine-grained control. It has also the potential to accelerate the learning process, e.g. where more extreme actions enable faster exploration. The approach is easy to use and yields competitive performance across a range of common torque-control benchmark tasks, as well as for generating PD-control targets within a high-fidelity simulation of the ANYmal quadruped, without extensive parameter tuning. While learning algorithms can always benefit from additional tuning, our approach increases robustness and helps to accelerate research in reinforcement learning for complex dynamic robots, in particular when there is no access to extensive computational resources.

**Acknowledgments**

Tim Seyde, Wilko Schwarting, Igor Gilitschenski, and Daniela Rus were supported in part by the Office of Naval Research (ONR) Grant N00014-18-1-2830, Qualcomm, and Toyota Research Institute (TRI). This article solely reflects the opinions and conclusions of its authors and not TRI, Toyota, or any other entity. We thank them for their support. The authors further would like to thank Lucas Liebenwein for assistance with cluster deployment, and acknowledge the MIT SuperCloud and Lincoln Laboratory Supercomputing Center for providing HPC resources. We would also like to thank the reviewers and program chairs for their helpful feedback and suggestions for improvement.

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
