# OpenReview forum: "Strength Through Diversity: Robust Behavior Learning via Mixture Policies"
_robot-learning.org/CoRL/2021/Conference — CoRL2021 Poster_

### Official Review · Reviewer_ADtN · 2021-07-23

**Originality:** Fair
**Technical Quality:** Good
**Clarity Of Presentation:** Good
**Impact:** 3

**Recommendation:**

Weak Accept: I recommend accepting the paper, but will not argue for my recommendation if the majority of other reviewers have a different opinion.

**Summary:**

This works aims to combat the need for extensive hyperparameter tuning by proposing a method that can utilize a single trial to combine different control designs conditional on the task. This reduces impact of hyperparameters and design decisions and shows more robustness on different simulated tasks.

The key contribution of this work seems to be reformulating the policy to be a mixture of experts style policy and then integrating this into the framework of Relative entropy regularized policy iteration

The key contributions of this work seem to be:

1. Introducing a mixture policy distribution where each component has different parameterization and potentially hyperparameters.
2. Modifying the KL regularized PI framework to add in the mixture policy.
3. Enabling optimization of mixed continuous and discrete distributions by introducing a pseudo density function.
4. Separating the constraints in KL regularized PI such that there is a separate constraint and corresponding lagrange multipliers per constraint.

There is a detailed empirical analysis combining different components and noting how the combination does better than individual pieces.

**Issues:**

The first and most glaring issue seems to be that this paper is entirely in simulation, with very little robotics. There is one domain which is the ANYmal robotics, but the rest is just standard mujoco tasks. This feels a bit underwhelming for a robotics conference, especially when the ANYmal task is just PD tracking.

In the introduction, the words “synergy” is used excessively and is not adding much value. The overall introduction is a bit too vague right now to add much value on what is actually happening, would correct this.

The paper describes the following phenomenon often: “This enables robust adaptation to a broad range of motion planning problems while reducing the necessity of manual parameter tuning and sequential experiment design” -> why is this?? I don’t see the clear reason why training a number of different models together makes this any  more robust? Could this be expanded more clearly? I’m assuming that it’s because multiple different hyperparameter models can be trained using the same data simultaneously and the best model is chosen automatically, but this could be spelled out a bit more for the reader.

Overall, the presentation of the paper makes it very hard to distinguish what is novel and introduced in this paper versus just adopted from past papers. Now it is perfectly ok to take a lot from past papers, it just needs to be made more clear in Section 4, and the key contributions need to be highlighted.

I think the methods section would also do well from a further discussion on why this framework helps with robustness and hyperparameter selections and such.

It’s also not clear why this objective would yield to actual specialization and not just collapse like many mixture of experts frameworks have. Can this be elaborated on in the paper?

Also a discussion on how to pick the different components would be very useful, currently that whole thing is left in the air. Also are the components trained with different hyperparameters?

It seems like the actual improvement in Fig 4 is quite marginal besides in the ANYMal domain. Can other domains be presented where this makes more of a difference? There is currently only one datapointn that this actually matters.

For the robustness plots, I think Fig 5 is actually a bit misleading. On closer inspection I can see that red is better than green on the quadruped, but the overall delta seems quite small and I was originally looking at just the error bars. I think more signposting should be done to indicate what to look for in the figures rather than “A is better than B”.

I think the comparisons against HOOF are a bit moot because the base RL algorithm is quite different. Can HOOF be combined with the same base RL algorithm here for Fig 6?

What about the robustness to hyperaparameters of the whole learning procedure and not the individual components??

Can Fig 7 be labeled with what each different line means properly? Which is the homogenous, heterogenous and the individual components?

I think Fig 7 and 8 have pretty good analysis but on individual components at a time. What happens when multiple are varied together? Also the difference in performance is not really going between working and not working altogether, but just small differences in return. Are there other domains where these robustness benefits are highlighted more clearly.


**Reviewer Expertise:**

Very good: Comprehensive knowledge of the area

**Strengths And Weaknesses:**

Strengths:
1. The approach seems simple
2. It is nice to be able to reuse data across multiple hyperparameters and architectures for controllers, especially very heterogeneous controller modes.
3. It is interesting that different controllers specialize to different parts of a trajectory.
4. The range of empirical results on actually trying different hyperparameters is a plus

Weaknesses:
1. Limited robotics experiments, results and discussion.
2. The improvement on the domains is somewhat marginal, especially over RHPO.
3. The figures and results are somewhat hard to parse without better signposting and labeling.


**Summary Of Recommendation:**

Overall I think the paper has an interesting premise, but I am recommending for a weak reject because of the results not being very much better than the baselines in many domains, a lack of robotics results and not a very clear novel contribution over prior work. I think a combination of multiple of these factors makes me recommend a weak rejection, but I am willing to reconsider given more experiments across other domains showing bigger improvements, significantly more clarity in the writing or more discussion or presentation of robotics results.

---

> ### Author Response · Authors · 2021-08-31
> **Response to Reviewer ADtN**
>
> We would like to thank the reviewer for their time and insightful comments. We are happy that the reviewer appreciates that our approach enables joint training of diverse policy parameterizations.
>
> >Limited robotics experiments, results and discussion, [while] the ANYmal task is just PD tracking.
>
> We have extended our evaluation to real-world inspired manipulation tasks from the Metaworld benchmarks, including bin picking, door opening, and using a hammer. Throughout, the NKCD HMP improves performance over RHPO. Regarding the ANYmal task, learning to generate position targets for a PD controller is a common formulation for learning real-world quadruped locomotion - we therefore believe the task constitutes an interesting domain.
>
> >The improvement on the domains is somewhat marginal, especially over RHPO.
>
> Our results on the Metaworld benchmark show additional improvement over RHPO. We note that RHPO is a strong baseline that was tuned to perform well across domains. The NKCD mixture is a particular instance of HMPs whose sub-policies vary along the distribution type dimension and our evaluation shows that training mixtures of diverse distribution types yields competitive performance even when individual components might fail on a task (see e.g. Figure 4, Kumaraswamy (K) on Humanoid). Our general objective is to show that we can train diverse sub-policies jointly, enabling the engineer to test various parameterizations while still performing competitively to strong baselines.
>
> >Are the components trained with different hyperparameters? What happens when multiple [components] are varied together?
>
> Yes, components are trained with different hyperparameters throughout and originally our experiments showed HMPs that varied their components along a single parameter dimension for ease of interpretation. We have now also added experiments for random component selection along multiple parameter axes. We show that random HMPs yield strong performance.
>
> >[Is training more robust] because multiple different hyperparameter models can be trained using the same data simultaneously and the best model is chosen automatically?
>
> Yes, robustness arises by not having to commit to a single policy parameterization and running multiple experiments to select the best, but letting the mixture automatically select within a single experiment.
>
> >It’s also not clear why this objective would yield to actual specialization and not just collapse.
>
> Specialization of components is not required and the policy may simply “collapse” onto the single best parameterization from its repertoire. However, emergent specialization based on task phases can be a by-product of this formulation as observed on the Cartpole and Cheetah tasks in Figures 2 & 3.
>
> >What is novel and introduced in this paper versus just adopted from past papers?
>
> Our HMP approach extends the MPO-based RHPO agent to the setting of diverse mixture policies and automatic hyperparameter selection. Our primary contribution is efficient training of diverse mixture policies that can vary along numerous parameter dimensions and in particular also distribution types. Furthermore, we enable joint training of both continuous and discrete policies on the same samples while decoupling each distribution type in the KL constraint. This perspective reduces hyperparameter optimization conducted over multiple experiments and instead enables the agent to try several combinations within a single experiment. This also enables the trained agent to self-select suitable controller designs for different phases of the same task, as observed e.g. on the Cartpole Swingup and Cheetah Run tasks in Figures 2 & 3.
>
> >Can HOOF be combined with the same base RL algorithm here for Fig 6?
>
> The closest method we found that optimizes hyperparameters within a single experiment is HOOF. The direct comparison is however rendered more complex as the authors state that 'HOOF can only optimise hyperparameters that directly affect the policy update', while our method optimises component specific parameters. These two sets have some overlap which is, however, incomplete. The quality of many other system parameters will affect the comparison such that one way to remain fair is to use the authors' original implementation. A combination between HOOF and our method is conceptually possible and requires non-trivial work. Therefore it presents a great direction for future work. We thank the reviewer for pointing in this direction.
>
> >Can Fig 7 be labeled with [...] the homogenous, heterogenous and the individual components?
>
> We have added descriptions of the line color to the caption.
>
> We thank the reviewer for their thorough feedback, and hope that we were able to clarify any open questions. We invite the reviewer to reconsider out submission based on the additional discussion and novel experimental evaluation, and to potentially adjust their score.

---

> > ### Comment · Reviewer_ADtN · 2021-09-02
> > **Response**
> >
> > I think the authors did a good job of addressing my concerns. and have showed new results on meta-world which are much more convincing, made a reasonable point about HOOF, done better signposting and labeling of experiments. Given the fact that they've addressed most of the weaknesses I've listed, I will raise my score to a weak accept.

---

### Official Review · Reviewer_PWdD · 2021-07-24

**Originality:** Fair
**Technical Quality:** Very Good
**Clarity Of Presentation:** Very Good
**Impact:** 3

**Recommendation:**

Weak Reject: I recommend rejecting the paper, but will not argue for my recommendation if the majority of other reviewers have a different opinion.

**Summary:**

The manuscript is proposing a method for hierarchical RL using a diverse set of sub-policies and training a mixture of them. The main idea is modeling a parametric policy as a mixture of sub-policies and simultaneously learning a non-parametric policy and fitting the parametric policy to the non-parametric one. The algorithm is later evaluated on various learning to control tasks.

**Issues:**

Related work covering Hierarchical RL and RL with options.

**Reviewer Expertise:**

Good: General knowledge of the area

**Strengths And Weaknesses:**

The consideration of the role of hyper-parameters and the hierarchical nature of the problem is appreciated. It is intuitive and fits the final application of RL for robotics better.

The proposed method is intuitive and mathematically sensible.

The results are very promising.

The major issue for me is the lack of proper coverage of related work. The paper clearly fits into the area of hierarchical RL as well as RL with options framework. However, the authors ignore this literature completely in their discussion of related work and empirical study.

The experimental study is not explained in detail. After reading the paper a few times, I am still confused about how many random trials are performed and what are the shaded areas in the figures. The baselines are also not justified. For example, the authors do not discuss why did they choose DDPG?

**Summary Of Recommendation:**

The method and the paper is sensible; however, I need a better justification for empirical study as well as an extended review of the relevant literature to recommend the paper for publication.

----

I thank the authors for their clarifications. It partially answers my concerns but the major issues remain. Hence, I will not change my score

---

> ### Author Response · Authors · 2021-08-31
> **Response to Reviewer PWdD**
>
> We would like to thank the reviewer for their time and positive feedback. We are excited that they appreciate our approach of hierarchical hyperparameter optimization and its good fit to robot control problems. In the following we would like to clarify open questions and remaining concerns.
>
> >The authors ignore [hierarchical RL as well as RL with options] completely in their discussion of related work and empirical study.
>
> We actually discuss several hierarchical RL and options approaches in the related works section in the third paragraph e.g. LL82. RHPO is an instance of a hierarchical agent which we compare against. We have modified the manuscript to improve clarity of these aspects.
>
> >I am still confused about how many random trials are performed and what are the shaded areas in the figures. [Also] the authors do not discuss why did they choose DDPG?
>
> We originally used 4 random seeds as reported in Appendix E, and have since extended the comparison with RHPO to 8 random seeds. We will also include this in the main body of the paper for improved clarity. The shaded areas represent one standard deviation and we have added this to the document. DDPG was chosen as a well-established reference on the Control Suite tasks - we will include a short motivation of this.
>
> Additionally, we have updated the manuscript to include evaluations on real-world inspired manipulation tasks from the Metaworld benchmark, and included performance of randomly sampled HMPs that vary along multiple parameter dimensions. We observe strong performance of HMPs throughout and believe this significantly strengthens our paper.
>
> We thank the reviewer for their insights and suggestions, and hope that we were able to address any open questions. We invite the reviewer to reconsider our submission based on the additional discussion and novel experiments, and to potentially adjust their score.

---

> ### Author Response · Authors · 2021-09-03
> **Clarification**
>
> Thank you for your reply and kind feedback! We are sorry about the delayed reply, but OpenReview did not notify us about updates to the reviews.
> Is there any particular work(s) from hierarchical RL or RL with options that the current discussion of [35]-[46] should be expanded by? Please feel free to suggest specific related work that we should add.

---

### Official Review · Reviewer_ghjV · 2021-07-25

**Originality:** Fair
**Technical Quality:** Fair
**Clarity Of Presentation:** Good
**Impact:** 4

**Recommendation:**

Weak Accept: I recommend accepting the paper, but will not argue for my recommendation if the majority of other reviewers have a different opinion.

**Summary:**

This paper presents a robust hyperparameter selection method that learns diverse sub-policies with various distribution types and parameterization. The authors use relative policy optimization [44] for the policy improvement step with modifications that use a sum of constraints consisting of decoupled components, each only varying along its respective parameter. The authors also use the piece-wise constant pseudo-density during policy evaluation that combines continuous action distribution and discrete action distribution. For empirical analysis, the authors evaluated the method on deepmind DM control tasks as well as OpenAI gym tasks comparing to prior gradient-free optimization approaches that showed that the method can achieve robust and sample-efficient learning.

**Issues:**

1. Extend the method to handle more diverse hyperparameters in RL such as critic and policy learning rates, architecture choices etc.
2. Perform evaluations on robotic manipulation tasks and also real-world tasks.
3. Provide some theoretical insight of the method.

**Reviewer Expertise:**

Good: General knowledge of the area

**Strengths And Weaknesses:**

The paper addresses an important problem in robot learning, which is the hyperparameter optimization issue. The proposed framework is neat since it can automatically train multiple sub-policies with different configurations in a hierarchical way, which is novel. The algorithm is simple yet effective in terms of the experimental results in simulated locomotion tasks. The paper is also well written and easy to understand.

However, I do have a few concerns, which I will discuss as follows.

First, the applicability of the proposed method seems a bit limited since the authors only consider different distribution types and parameterization of the policy in RL for hyperparameter optimization, which only covers a very small portion of the hyperparameters that matter in RL. The experimental results also only focus on simulated locomotion tasks, which seem not comprehensive. I think it would be important to extend the method to handle a wider range of hyperparameters in RL, e.g. learning rate, architecture design, etc. It is also pivotal to test the method on robotic manipulation tasks and also real-world tasks, which are more realistic for robot learning.

Moreover, the algorithm appears a bit incremental, it seems to be almost a direct adaptation of the prior work [44] to the setting mixture of policies. The theoretical insight of such a direct adaptation is also missing.

================================

After reading the rebuttal, I appreciate that the authors conduct additional more realistic robotic experiments and also clarified my concerns on novelty of the paper. However, my concern on automatically selecting other hyperparameters still remains. The authors claim that their approach can automatically select hyperparameters and only show the use case for the distribution types. I think such a claim cannot be fully substantiated in the paper, therefore limiting the contribution of the paper. I will keep my score.

===============================

More updates: thank you for pointing me to the additional experiments on randomly sampled hyperparameters! I've increased my score to a 6. Though I still have a few questions as follows. It seems that NKCD does not outperform RHPO and is even a bit worse in several domains. It is not also clear to me if RHPO is a reasonable baseline for varying other hyperparameters. The authors should perhaps compare to the oracle, which is DDPG tuned with each of the varying hyperparameter.

**Summary Of Recommendation:**

Given my two points on weaknesses of the proposed method discussed in the Strengths And Weaknesses section, I would vote for a weak reject.

---

> ### Author Response · Authors · 2021-08-31
> **Response to Reviewer ghjV**
>
> We would like to thank the reviewer for their time and positive feedback. We appreciate them pointing out the importance of automatic hyperparameter selection in RL for robotics and the novelty of our approach. In the following, we will try to clarify open questions and address remaining concerns.
>
> >The proposed framework is neat since it can automatically train multiple sub-policies with different configurations in a hierarchical way, which is novel. The algorithm is simple yet effective in terms of the experimental results in simulated locomotion tasks. The paper is also well written and easy to understand.
>
> We would like to thank the reviewer for their positive feedback and commending the effectiveness of our method.
>
> >It is [...] pivotal to test the method on robotic manipulation tasks and also real-world tasks.
>
> We have extended our evaluation to robot manipulation tasks from the Metaworld benchmarking suite and include real-world inspired tasks such as bin picking, opening a door and using a hammer. Throughout, we observe improved performance of our NKCD mixture over RHPO. It should be noted that NKCD is just one instance of our Hyperparameter Mixture Policies (HMPs) that varies sub-policies along the distribution dimension. This serves to underline HMPs’ robust performance even under individual component failure as observed in Figure 4 (top) for e.g. the Kumaraswamy component (K) on Humanoid Walk.
>
> >I think it would be important to extend the method to handle a wider range of hyperparameters in RL, e.g. learning rate, architecture design, etc.
>
> We currently primarily focus on diversity in distribution types and parameterizations, while also investigating different feedforward network architectures and activations. The combination of different distribution types within a single hierarchical policy is not common in RL and we show that good performance can be achieved with non-Gaussian components, which is in part surprising considering the predominant use of Gaussian policies throughout the literature. Furthermore, layer architecture and activations can have severe effects on performance as investigated in “Deep Reinforcement Learning that Matters” by Henderson et al. (2018), [6].
>
> >The algorithm [...] seems to be almost a direct adaptation of the prior work [44] to the setting mixture of policies. The theoretical insight of such a direct adaptation is also missing.
>
> Our HMP approach extends the MPO-based RHPO agent to the setting of diverse mixture policies and automatic hyperparameter selection. Our primary contribution is efficient training of diverse mixture policies that can vary along numerous parameter dimensions, while enabling training of continuous and discrete policies jointly on the same sample data. This perspective reduces hyperparameter optimization conducted over multiple experiments and instead enables the agent to try several combinations within a single experiment. This also enables the trained agent to self-select suitable controller designs for different phases of the same task, as observed e.g. on the Cartpole Swingup and Cheetah Run tasks in Figures 2 & 3.
>
> We thank the reviewer for their insights and suggestions, and hope that we were able to address any open questions. We invite the reviewer to reconsider our submission based on the additional discussion and novel experiments, and to potentially adjust their score.

---

> > ### Author Response · Authors · 2021-09-03
> > **Clarification**
> >
> > Thank you for your reply and kind feedback! We are sorry about the delayed reply, but OpenReview did not notify us about updates to the reviews.
> > We would like to clarify that the new random component sampling experiment in Figure 5 randomly selects not only distribution types but also the layer architecture, initializations and activation functions. We are happy to discuss further!

---

> > > ### Author Response · Authors · 2021-09-04
> > > **Additional clarification**
> > >
> > > Thank you very much for your additional feedback! Our NKCD HMP does yield improved performance over RHPO throughout the new manipulation tasks in row 3 of Figure 4. We initially chose a configuration for RHPO that yields very high-performance across the DeepMind Control suite tasks and ANYmal in row 2 of Figure 4 to highlight strong performance of HMPs that is robust to potentially sub-optimal individual sub-policies (e.g. Figure 4, row 1). The manipulation domains can therefore in a sense be interpreted as OOD tasks for the agents' hyperparameters, where the NKCD HMP continues to yield highly competitive performance. This behavior nicely underlines that HMPs can yield robust performance without additional fine-tuning. We agree that performance of varying hyperparameters of RHPO/DDPG would be an insightful ablation, and would happily add it to the appendix of a final version.

---

### Official Review · Reviewer_1NVm · 2021-07-30

**Originality:** Good
**Technical Quality:** Good
**Clarity Of Presentation:** Fair
**Impact:** 4

**Recommendation:**

Weak Accept: I recommend accepting the paper, but will not argue for my recommendation if the majority of other reviewers have a different opinion.

**Summary:**

This paper proposes to learn a “meta-policy” that selects the best policy among a set of policies, depending on the state. The idea of the authors is that each sub-policy could have a different structure or be trained with different hyper-parameters: the meta-policy can therefore be viewed as an hyper-parameter selection algorithm (we do not have to select the hyper-parameters, the meta-policy does it). The authors test their idea with the Deep Mind Control Suite and an experiment with the ANYmal simulation.

**Issues:**

- The results critically depends on the choice of the sub-policies, which are hand-picked.

- Only 4 seeds were used, which is too little to perform any statistics and draw any conclusion in empirical work (most statisticians would say "at least 20, although more is almost always better). First, the number of seeds should be in the caption of each result figures (so that we have the elements to interpret the figures). Second, are the figures representing the mean/standard deviation? mean/standard error? median/95% CI? This should also be in the caption.


- The authors argue that their approach is faster/more data-efficient than alternatives (e.g., grid search); is it because all the competing sub-policies use the same samples? it is never clearly stated that/if all the sub-policies are trained together with the same samples.

- The number of samples might be lower than learning each sub-policy independently, but all the rest of the computation (e.g., activating the networks) is the same as learning each of them independently; this should at least be discussed.

- How is 4.1 different from the ref. 44 and from any other actor-critic algorithm? what is specific to the hierarchical nature of the problem here? The authors should highlight the critical novelties/changes here.

### Other issues
- The experiments on architecture variations (section 5.3, figure 7b) are probably not statistically significant (hard to say since the authors do not perform any statistical analysis). Please remove the claim that the proposed method is “slightly better” or provide some p-values.
- There is an important literature about “quality diversity” that aims at finding automatically a diverse set of policies, which seems to be very relevant for this paper (but overlooked by the authors). For instance, check:
  * Duarte M, Gomes J, Oliveira SM, Christensen AL. Evolution of repertoire-based control for robots with complex locomotor systems. IEEE Transactions on Evolutionary Computation. 2017 Jun 30;22(2):314-28.
   * Nilsson O, Cully A. Policy gradient assisted MAP-Elites. Proc. of GECCO 2021.


**Reviewer Expertise:**

Fair: Some knowledge of the area

**Strengths And Weaknesses:**


### Strengths
- Interesting idea
- A lot of different experiments are reported

### Weaknesses
- the sub-policies have to be chosen and might not be actually diverse; a simple baseline/variant would have been to use a set of sub-policies with random hyper-parameters (so that we do not have to choose, like in standard hyper-parameter tuning/parameter control algorithms)
- many important parts of the algorithms are unclear to me; in particular, it is never clearly stated that/if the sub-policies are trained simultaneously
- only 4 replicates (!), which is far from being enough to perform any statistics
- no real robotics experiments and most of the experiments (except maybe ANYmal) are actually not close to a robot
- the authors mention related work that does automatic parameter control/tuning, but there is no empirical comparison to this kind of technique
- no statistical analysis (p-value, etc.): on many plots, the variances seem to overlap, which strongly suggest that most differences are not statistically significant


**Summary Of Recommendation:**

Overall, this is a promising paper that describes a large amount of work. However, while each paragraph is well written, it found it difficult to follow the main story and the main ideas.

In addition, I think the paper would be much stronger if the sub-policies were automatically generated (e.g., random hyper-parameters, grid search-like, etc.) instead of hand-picked.

I initially suggested a "weak accept" but I moved to "weak reject" when I finally found in the supplementary material that only 4 seeds were used. This is too little to conclude anything from the empirical comparisons, especially when many results look very similar.

---

> ### Author Response · Authors · 2021-08-31
> **Response to Reviewer 1NVm**
>
> We would like to thank the reviewer for their time and extensive comments. We are happy that the reviewer underlines the promise of our approach to automatic hyperparameter selection. In the following, we will try to clarify open questions and address remaining concerns.
>
> >Most of the experiments (except maybe ANYmal) are actually not close to a robot.
>
> We provide additional manipulation experiments on real-world inspired tasks from the Metaworld benchmark in Figure 4. The tasks include bin picking, opening a door, and using a hammer among others. Here, we observe strong performance of our NKCD Hyperparameter Mixture Policy (HMP) compared to the RHPO agent.
>
> >Only 4 seeds were used, which is too little [ ...]
>
> We have extended the comparison with RHPO to 8 seeds and provide additional manipulation experiments. Generally, our focus is less on quantitative performance but the ability to successfully train various policy parameterizations simultaneously and even recover performance that is competitive with tuned state-of-the-art agents. We show this across various parameter axes and environments, and believe that 4 seeds can provide valuable qualitative insights, while 5 seeds are commonly used in RL to make quantitative claims.
>
> >Are the figures representing the mean/standard deviation?
>
> Yes, thank you for pointing this out, we have highlighted this in the text.
>
> >The sub-policies have to be chosen and might not be actually diverse [...] I think the paper would be much stronger if the sub-policies were automatically generated (e.g., random hyper-parameters, grid search-like, etc.)
>
> The sub-policies do not need to be chosen explicitly and we only did so to provide controlled experiments along a single parameter dimension, e.g. distribution type in the RHPO comparison. We have added novel experiments that randomly sample 10 components from a parameter hyper-space and show that this approach yields high-performance policies in Figure 5. However, initial component choice may be an opportunity for the engineer to inject structural priors as in Figures 2 & 3.
>
> >It is never clearly stated that/if the sub-policies are trained simultaneously [...] with the same samples.
>
> The sub-policies are trained simultaneously (see e.g. L45, LL160). We will state this more prominently at the beginning of Sec. 4 for improved clarity. Simultaneous training is the primary reasons for reduced sample complexity over the mentioned grid search approaches, as multiple policy parameterizations can be evaluated jointly within a single experiment.
>
> >The authors mention related work that does automatic parameter control/tuning, but there is no empirical comparison.
>
> The discussed methods commonly rely on either parallel or sequential experiments thereby significantly increasing the number of interactions. The closest method we found that optimizes hyperparameters within a single experiment is HOOF and we provide performance comparisons on the Open AI Gym environments they originally used.
>
> >The number of samples might be lower [...], but all the rest of the computation is the same as learning each of [the sub-policies] independently.
>
> Training a mixture of N sub-policies does indeed increase required memory over training a single policy, although memory requirements will scale sub-linearly when comparing a mixture of N sub-policies to N runs of a single policy. Here, our main concern is sample-efficiency which our HMPs can significantly reduce over parallel or sequential evaluation approaches.
>
> >How is 4.1 different from the ref. 44 and from any other actor-critic algorithm?
>
> The computations presented in Section 4.1 follow the Expectation-Maximization type algorithm used by MPO and RHPO, which differs from conventional actor-critic algorithms by optimizing towards an intermediate non-parametric policy and not requiring noisy gradient estimation via likelihood ratios [45] or reparametrization [46]. A critical difference is that our formulation allows for simultaneously training heterogeneous sub-policies differing along various parameter dimensions (e.g. distribution type, layer architecture, activations, initializations) while allowing joint training of continuous and discrete distributions within the same mixture.
>
> >There is an important literature about “quality diversity” [...] which seems to be very relevant for this paper
>
> We thank the reviewer for referencing the paper by Duarte et a (2018) and the very recent work by Nilsson et. al (2021). We have added these quality diversity methods to our discussion of diverse skill learning approaches in the third paragraph of the related works.
>
> We thank the reviewer for their extensive feedback, and hope that we were able to clarify any open questions and address remaining concerns. We invite the reviewer to reconsider our submission based on the additional discussion and experiments, and to potentially adjust their score.

---

> > ### Comment · Reviewer_1NVm · 2021-09-02
> > **Updated version**
> >
> > Thank you for the updated version. I have moved my recommendation to "weak accept" given the improvements (in particular, the higher number of seeds).
> >
> > Re-reading the paper, I think the version based on random parameters for the sub-policies is the most interesting one since there is no prior on the hyper-parameters. I encourage the authors to test this approach more.
> >
> > Regarding statistics (and no, 5 seeds is usually not enough, even if many papers do it...), please have a look at:
> >
> > - https://arxiv.org/abs/2108.13264
> > - https://arxiv.org/abs/1806.08295
> >
> > This will be very useful for your next paper (or a new version of this one).

---

### Meta-Review · Area_Chair_Pe8h · 2021-08-11

**Recommendation:** Accept (Poster)
**Confidence:** 4

**Metareview:**

After the discussion and rebuttal from the authors providing additional evidence for their method, some of the reviewers updated their scores to accept the paper. The remaining points of criticism are mostly regarding the related work section, which can be fixed in the final version of the paper. I encourage the authors to carefully read the reviews and address the points raised by the reviewers in the final version of the paper.

---

> ### Author Response · Authors · 2021-08-31
> **Updated manuscript and main changes**
>
> We would like to thank the reviewers and the AC for their time and valuable feedback. We have updated our manuscript with the following main changes:
>
> - Extension of the comparison with RHPO to 8 seeds (Figure 4)
> - Evaluation on real-world inspired manipulation domains from the Metaworld benchmark showing improved performance over RHPO (Figure 4)
> - Evaluation of randomly generated sub-policies against RHPO (Figure 5)
> - Improved clarity of writing in several places
>
> We would like to emphasize that HMPs offer a framework for simultaneously training policies with various distribution types and parameterizations. While individual components may fail to solve a task, the mixture yields robust performance that is competitive with the tuned baseline RHPO agent, even improving upon it in several domains. Generally, the hyperparameters we consider were inspired by P. Henderson et al. “Deep reinforcement learning that matters (2018) ([6]), where they were shown to have a significant impact.

---

### Decision · Program_Chairs · 2021-09-13

**Decision:**

Accept (Poster)

**Comment:**

After the discussion and rebuttal from the authors providing additional evidence for their method, some of the reviewers updated their scores to accept the paper. The remaining points of criticism are mostly regarding the related work section, which can be fixed in the final version of the paper. I encourage the authors to carefully read the reviews and address the points raised by the reviewers in the final version of the paper.